# Interplay of Vitamin D and SIRT1 in Tissue-Specific Metabolism—Potential Roles in Prevention and Treatment of Non-Communicable Diseases Including Cancer

**DOI:** 10.3390/ijms24076154

**Published:** 2023-03-24

**Authors:** Zsuzsanna Nemeth, Attila Patonai, Laura Simon-Szabó, István Takács

**Affiliations:** 1Department of Internal Medicine and Oncology, Semmelweis University, Koranyi S. u 2/a, 1083 Budapest, Hungary; 2Department of Surgery, Transplantation and Gastroenterology, Semmelweis University, Ulloi u. 78, 1082 Budapest, Hungary; 3Department of Molecular Biology, Semmelweis University, Tuzolto u. 37-47, 1094 Budapest, Hungary

**Keywords:** vitamin D, SIRT1, metabolism, epigenetics, signaling pathways, prevention and treatment

## Abstract

The importance of the prevention and control of non-communicable diseases, including obesity, metabolic syndrome, type 2 diabetes, cardiovascular diseases, and cancer, is increasing as a requirement of the aging population in developed countries and the sustainability of healthcare. Similarly, the 2013–2030 action plan of the WHO for the prevention and control of non-communicable diseases seeks these achievements. Adequate lifestyle changes, alone or with the necessary treatments, could reduce the risk of mortality or the deterioration of quality of life. In our recent work, we summarized the role of two central factors, i.e., appropriate levels of vitamin D and SIRT1, which are connected to adequate lifestyles with beneficial effects on the prevention and control of non-communicable diseases. Both of these factors have received increased attention in relation to the COVID-19 pandemic as they both take part in regulation of the main metabolic processes, i.e., lipid/glucose/energy homeostasis, oxidative stress, redox balance, and cell fate, as well as in the healthy regulation of the immune system. Vitamin D and SIRT1 have direct and indirect influence of the regulation of transcription and epigenetic changes and are related to cytoplasmic signaling pathways such as PLC/DAG/IP3/PKC/MAPK, MEK/Erk, insulin/mTOR/cell growth, proliferation; leptin/PI3K-Akt-mTORC1, Akt/NFĸB/COX-2, NFĸB/TNFα, IL-6, IL-8, IL-1β, and AMPK/PGC-1α/GLUT4, among others. Through their proper regulation, they maintain normal body weight, lipid profile, insulin secretion and sensitivity, balance between the pro- and anti-inflammatory processes under normal conditions and infections, maintain endothelial health; balance cell differentiation, proliferation, and fate; and balance the circadian rhythm of the cellular metabolism. The role of these two molecules is interconnected in the molecular network, and they regulate each other in several layers of the homeostasis of energy and the cellular metabolism. Both have a central role in the maintenance of healthy and balanced immune regulation and redox reactions; therefore, they could constitute promising targets either for prevention or as complementary therapies to achieve a better quality of life, at any age, for healthy people and patients under chronic conditions.

## 1. Introduction

Prevention and control of non-communicable diseases, including obesity, metabolic syndrome, type 2 diabetes, cardiovascular diseases and cancer, which possess a particular mutually inclusive relationship, should be as important part of the health care as the treatment of these diseases. Although it is already well known, from the example of smoking habits, that people often do not want to effectuate changes in their lives in order to improve their health, the research and transfer of scientific knowledge into practice and the education of patients must not be impeded and give up. Older age should not necessarily be associated with diseases, as a high percentage of non-communicable diseases could be preventable or diagnosed in an early state where adequate lifestyle changes, alone or with necessary treatments, could reduce the risk of mortality or a deterioration in quality of life. The expressions such as “aging population” and “sustainable health care”, which are used more often recently, both highlight the necessity to reduce the incidence of unnecessary disease development. Similarly, the 2013–2030 action plan of the WHO for the prevention and control of non-communicable diseases seeks these achievements. 

In relation to the abovementioned, wide scientific field, in this review we aim to select and detail the role of only two “molecular participants”: vitamin D and sirtuin 1 (SIRT1). Both have received increased attention in relation to the COVID-19 pandemic [1,2], as they take part in the regulation of the main metabolic processes, i.e., lipid/glucose/energy homeostasis [3,4,5,6], oxidative stress [7,8], redox balance [9], and cell fate [10], as well as in the healthy regulation of the immune system [11,12]. Additionally, they are interconnected in several layers of cellular mechanisms and in the homeostasis included in the epigenetic regulation of transcription [13], metabolic and energy homeostasis [14], balance of inflammatory processes [12,15], or processes which influence cardiovascular health [16,17].

It is already known from several studies that vitamin D supplementation can significantly reduce all-cause mortality and cardiovascular disease (CVD) mortality [18,19,20,21,22,23]. Additionally, it is acknowledged that deficiency of vitamin D is associated with several diseases, e.g., obesity [24], metabolic syndrome [25], immune dysfunction [26], insulin resistance [27], cardiovascular disease, renin-angiotensin system/ventricular hypertrophy/stroke [28], Crohn’s disease/ulcerative colitis/inflammatory bowel disease [29], systemic sclerosis [30], cancer [31,32], and dementia/depression/schizophrenia/ADHD [33]. Many of these are associated with a reduced level of SIRT1 as well [34,35]. Vitamin D levels may influence the risk of infections in kidney and liver transplant recipients [36,37] and the risk of acute rejection [38,39] and of new-onset diabetes after kidney transplantation [40].

As the length of a review is limited, here we aim only to highlight some important interconnecting fields of the actions of vitamin D and SIRT1. Those who are interested in greater details and would like to get to know more overlapping areas can find information in the references.

## 2. The Role of Vitamin D and SIRT1 in Healthy Metabolic Regulation of Different Tissues

In this section, we will detail the role of vitamin D and, briefly, SIRT1 in healthy metabolic regulation. Vitamin D is discussed here in more detail and length since the role of SIRT1 had been already published in our previous work, where it is easily accessible for those who interested in [34,35].

Vitamin D is a well-known hormone regulating bone metabolism, but there is growing evidence of its non-classical physiological roles; however, much remains to be learned about the underlying molecular mechanism of these effects [41,42]. This hormone has a much wider role in the homeostasis and cellular metabolism of the human body than simply the regulation of the calcium and phosphate level (Figure 1).

### 2.1. Modes of Action of Vitamin D

Two different modes of action of vitamin D are known so far. Its genetic effects are dependent of its nuclear receptor (VDR), which is expressed in many tissues and cell types and modulates thousands of genomic loci via chromatin changes and expression of hundreds of primary target genes. It takes a few hours in an in vitro experiment to observe the physiological consequences after treatments with an amount greater than the physiological concentration of this nuclear hormone. In contrast, its non-genetic effects occur within seconds to minutes, with or without a VDR, but without effecting gene transcription [43]. Of note, a membrane-associated VDR is also recognized; this interacts with CAV1, SRC, and PDIA3, opens membrane-signaling cascades with PLC, DAG/IP3, PKC, and MAPK, and may downregulate Wnt, SSh and Notch signaling [44]. However, the physiology of the two actions largely overlaps [43].

The effects of vitamin D on epigenetic regulation are led by changes in the chromatin accessibility of VDRs. This is supported by the DNA binding of its enhancers, i.e., partner pioneer factor (PU.1) [45,46], and via vitamin D-sensitive CTCF sites. These latter affect the 3D structure of chromatin through activation of some 600 of 3000 topologically associated domains (TADs) [47,48], where vitamin D target genes are located [32].

### 2.2. Role of Vitamin D in Lipid Metabolism

In relation to lipid metabolism, vitamin D supplementation significantly decreased body mass index (BMI) and body fat mass in adult patients with type 2 diabetes in a randomized, placebo-controlled, double-blind trial, while significantly increasing the level of adiponectin. Vitamin D supplementation also significantly enhanced levels of SIRT1 and SIRT6; however, the enhancement of the latter was more pronounced when vitamin D was applied with Ca^2+^ [49]. SIRT6 plays an important role in glucose homeostasis by increasing insulin secretion parallel with inhibition of gluconeogenesis and lipogenesis and suppression of obesity-induced inflammation and insulin resistance [50,51]. Adiponectin, secreted mainly by adipose tissue, but also by the skeletal muscle, acts as a hormone taking part in glucose and lipid metabolism as well as insulin sensitivity [52]. Adiponectin decreases the level of triglycerides (TGs) while increasing the level of high-density lipoprotein (HDL-C) via increased hepatic production of apo-AI and ATP-binding cassette transporter A1 (ABCA1)—a known player in HDL assembly [53]. Adiponectin also increases fatty acid oxidation, reduces hepatic glucose production, and increases glucose uptake and utilization in skeletal muscle [49]. These processes, with their activating effects on adenosine monophosphate-activated protein kinase (AMPK) and peroxisome proliferator-activated receptor-α (PPAR-α), improve insulin resistance in the liver and skeletal muscle [54]. Adiponectin could reduce inflammatory reactions as well, i.e., it may inhibit NFĸB signaling, and decrease CRP level and TNFα secretion from activated macrophages [55]. Functions and phenotypes of macrophage and dendritic cells are also modulated by adiponectin via stimulation of IL-10 and IL1-RA as part of adiponectin’s anti-inflammatory property [56]. It is not surprising, considering all the abovementioned effects, that adiponectin has an antiatherogenic role. AMPK partly produces nitric oxide (NO), which supports endothelial health by relaxing endothelial smooth muscle and lowering blood pressure, and, on the other hand, protects against cardiac hypertrophy as a result of SIRT1 upregulation, PPAR-α activation, and increased fat oxidation, while simultaneously reducing levels of inflammation [34,35].

### 2.3. Role of Vitamin D in Cardiovascular System

Vitamin D plays direct role in the health of the cardiovascular system as well through its receptor, VDR [57], the expression of which was detected both in endothelial and smooth muscle cells [58,59] as well as in the human heart [60]. Endothelial cells participate in the regulation of blood vessel relaxation and contraction by endothelial-derived factors and by nitric oxide (NO) production. Indeed, both 1,25(OH)_2_D_3_ and its analogs are able to significantly reduce vascular tone by decreased production of contracting factor—as the result of a reduced calcium influx into the endothelial cells—and by increasing the amount of endothelial NO [61,62]. Vitamin D regulates myocardial maturation and function, and downregulates the renin-angiotensin-aldosterone axis, thus contributing to reduction of hypertrophic cardiovascular remodeling, myocardial fibrosis, and hypertension [63]. Vitamin D affects vasculature by regulating calcium homeostasis as well [57]. In this context, however, not only vitamin D level, but the dynamic interaction between the parathyroid gland, kidney, bone, vitamin D metabolites, vitamin D-binding protein (DBP), SIRT1-Axl, and miR-34a level are all important. In healthy individuals, these interactions control, minute to minute, the calcium homeostasis and are particularly responsive to vitamin D and calcium intake [63,64]. Moreover, the optimal physiological parameters of these interactions change with growth, maturation, and aging. Thus, to prevent ectopic mineralization (i.e., vascular calcification or tissue calcinosis), it should not be assumed that calcinosis is merely a simple consequence of elevated calcium phosphate products due to vitamin D levels, but should be understood as an actively regulated, complex process where substantial heterogeneity is observed [65,66]. Additionally, this process is influenced and followed by local inflammation and oxidative stress [67,68,69]. However, based on atherosclerosis models, this can also be prevented by a normal level of vitamin D, which limits foam cell formation and macrophage activation with vascular inflammation [70], similarly to VDR analogs which also reduce inflammation [67]. Moreover, in uremic rats, these VDR analogs diminish cardiovascular oxidative stress [69]. Furthermore, 1,25(OH)_2_D_3_ inhibits endothelial cell proliferation and angiogenesis in a dose-dependent manner, which is assumed by less vascularized tumors in 1,25(OH)_2_D_3_-treated mice [58,71]. However, the active vitamin D hormone has a promitogenic effect as well, increasing the proliferation rate of aortic smooth muscle cells, but it is VEGF-dependent, and VEGF receptor antagonists could blunt this process [72,73].

### 2.4. Role of Vitamin D in the Immune System

VDR evolved some 550 million years ago in a boneless vertebrate where, at that time, its evolutionary function was to control the metabolism in order to support the immune system of ancestral vertebrates with energy [74]. This was confirmed based on the genome-wide analysis of vitamin D signaling on the most investigated cell types of the immune system, i.e., THP-1/monocyte/ and PBMCs. It is supposed that VDR and its ligand first specialized in the modulation of innate and adaptive immunity [75,76] before they took on the task of regulating bone metabolism [77]. VDR is expressed by a plethora of immune cells including monocyte/macrophages, dendritic cells, neutrophils, and T and B lymphocytes [78,79]. Therefore, these cells are a potential target for the prevention of immune-related diseases such as multiple sclerosis and rheumatoid arthritis [80,81,82]. An amount of 15 [77], 10 [83], or 19 [84] key vitamin D target genes were found and classified according to their primary functions in immune-related cells depending on selected datasets and analysis (Table 1). Most of the proteins encoded by these genes are located in the plasma membrane; some of these are secreted, and few of them act in the nucleus or in the cytoplasm (Table 1). Their main functions in the immune system were defined as: 1. autoimmunity, 2. infection in general, 3. acute response to infection [77], or as: 1. immune response, 2. metabolism and transport, 3. proliferation, differentiation, and apoptosis [83], or as: 1. neutrophil activation, 2. positive regulation of TNF production, 3. inflammatory response, 4. neutrophil degranulation, 5. negative regulation of T cell proliferation, 6. positive regulation of cytokine secretion [84]. Additionally, signaling pathway impact analysis (SPIA) resulted in the identification of seven common pathways that are primarily based on the above indicated target genes: (a) Th1 and Th2 cell differentiation, (b) Th17 cell differentiation, (c) antigen processing and presentation, (d) inflammatory bowel diseases, (e) tuberculosis, (f) leishmaniasis, and (g) toxoplasmosis [83]. Moreover, downregulation of HLA-DRA and HLA-DRB1 were prominently involved in all the seven pathways, while the upregulation of LAMB3 played a role in toxoplasmosis [83]. Vitamin D reduces chronic inflammation by inhibition of the Akt/NFĸB/COX-2 pathway [85,86] and through Nrf2 activation [87]. Through the action of adiponectin, Vitamin D inhibits NFκB signaling and decreases inflammatory cytokines, such as TNFα, and CRP level; in addition, Vitamin D possesses anti-inflammatory properties through stimulation of IL-10 and IL1-RA [55,56]. Additionally, vitamin D signaling in macrophages induces the catabolism of the branched-chain-amino acid (BCAA) in a cell-specific manner [42]. The level of BCAA is an indicator of the metabolic status, the level of protein synthesis and the degree of autophagy, as well as other metabolic pathways regulated by mTOR in myeloid cells [88]. Downregulation of mTOR leads to the enhancement of autophagy, which plays a role in controlling intracellular pathogens by macrophages [42]. These data suggest that vitamin D has a wide range and complex effect on the regulation of a healthy immune system.

### 2.5. Role of Vitamin D in Redox Homeostasis

In redox homeostasis, vitamin D and plant polyphenols, i.e., green tea catechins, hydroxytyrosol (HT), curcumin, and NAD/NADH-SIRT1-inducer resveratrol in a low dose, can activate antioxidant signaling pathways to maintain cellular redox homeostasis by the upregulation of cytoprotective genes (i.e., heat shock protein 70/Hsp70/, heme oxygenase-1/HO-1/, glutathione redox system, SIRT1, NAD(P)H:quinone oxidoreductase/NQO1/, γ-glutanylcysteine synthetase/γ-GCS, etc.), important players in the redox stress response and autophagy mechanism [87]. In this context, mutagenic agents, such as environmental pollutants, radiation, or dietary mutagens [89,90], as well as chronic metabolic conditions [34,35,91], may result in the formation of reactive oxygen species (ROS) and reactive nitrogen species (RNS), which then undergo chemical reactions with cellular proteins, lipids, and DNA contents, inducing carcinogenesis [87,92]. Vitamin D and polyphenols—in addition to their scavenging effects on ROS/RNS, and similarly to ROS itself—can activate Nrf2 transcription factor, which regulates the expression of antioxidant enzymes, detoxifying factors, antiapoptotic proteins, and drug transporters [93]. Through this process, Vitamin D and polyphenols protect normal cells from carcinogenesis and are able to suppress the progression of cancer cells and sensitize them to therapy [87,94,95]. The antioxidant effect of vitamin D was demonstrated in hyperglycemia-induced renal injury [95]. These data suggests that plant vitamin D and polyphenols have application potential in the prevention and treatment of the aforementioned diseases, including cancers [87].

Although there are several other areas where vitamin D acts in the human body, in relation to SIRT1 action and the possible double action in prevention of non-communicable diseases, those detailed are the most important processes, as these have a broad influence on other cellular and metabolic processes and on homeostasis.

### 2.6. Role of SIRT1 in Healthy Metabolic Regulation

In our previous review, the specific role of SIRT1 in the healthy metabolic regulation of different tissues was already detailed. We also highlighted the feasibility of lifestyle modifications (appropriate physical activity, calorie intake, dietary intake, circadian rhythm, emotional/cognitive states) to achieve higher levels of SIRT1 for optimal tissue-specific functions [34,35]. Briefly, SIRT1 is significantly increased by calorie restriction, changes in the composition of one’s diet, appropriate type and intensity of exercises, and by positive mental function in both healthy and diseased patients. These lifestyle changes, correlated with increased SIRT1, were followed by beneficial physiological changes such as significantly decreased body weight, visceral fat, BMI, triglicerid, LDL, insulin, glycated haemoglobin, and leptin level, while significantly increased the levels of NOx, HDL, FFA, the insulin sensitivity, the level and activity of AMPK, PPARγ and PGC-1α level, and mitochondrial biogenesis [96,97]. These resulted in significant changes in the function of the immune system, namely, a significant decrease in the TLR4 proinflammatory pathway and in IFNγ, IL-1β, IL-6, IL-8, and TNFα levels [98]. In addition, significant decreases were observed in oxidative stress level, insulin/mTOR/cell growth and proliferation, and in leptin/PI3K-Akt-mTORC1 pathways, which are implicated in obesity-related conditions such as diabetes, cardiovascular diseases, and cancer [99,100]. In addition, these lifestyle changes significantly reduced activation of the inflammation-related NFκB pathway and expression of MMP-9 [101,102]. Additionally, resveratrol, a dietary polyphenol, induces SIRT1 expression and, by activating signaling pathways and transcription factors, regulates cell cycle, apoptosis, and angiogenesis in immune cells [103,104,105]. This dietary polyphenol reduces the level of proinflammatory cytokines (TNF-α, IL-1β), enzymes (iNOS, COX-2), and activation of signaling pathways (NFκB) [106]. The anti-inflammatory effects of resveratrol suggest its application as a complementary therapy for immune-mediated diseases [104]. Additionally, resveratrol inhibits initiation, promotion, and progression of cancer development [107,108]. In redox the mechanism, it shows scavenging capacity, by increasing levels of NOS and SIRT1, and improves mitochondrial function [109]. Moreover, in cancer cells, resveratrol can promote ROS production for cell cycle arrest and apoptosis [110,111]. On the other hand, it is protective to neurons, as demonstrated in Alzheimer’s and Parkinson’s models [112].

### 2.7. Connection and Importance of Circadian Rhythm in SIRT1-Regulated Metabolism

As we described in our previous review, the circadian rhythm is the basis of all diurnal behaviors and energy/metabolic processes [34,35,113]. Circadian control of cellular metabolism has two main clocks: 1. the master or central clock in the suprachiasmatic nucleus (SCN) receives direct input from the retina (light/dark cycle), and 2. the peripheral clock includes feeding–fasting cycles, temperature cycles, and chemical inducers such as dexametasone [114]. The two main clocks can control/influence each other. It has been reported that circadian alterations are frequent in metabolic syndrome, cancers, and mood disorders [115]. According to descriptions in the literature, smoking is able to disrupt circadian rhythm, decreases SIRT1 expression, and is strongly associated with metabolic imbalances and inflammation [116,117]. SIRT1 deacetylates two central components of the circadian clock (BMAL1 and PER2) in the liver and, thus, controls liver-specific circadian regulation of metabolic processes and, through PGC-1α activation, the metabolic and energy pathways connected to transcriptional output [97,118].

## 3. Vitamin D and SIRT1 in Relation to Non-Communicable Diseases Including Cancer

In addition to the skin, both the liver and the kidney are key organs in vitamin D metabolism. The inactive vitamin D3 (cholecalciferol), which is formed in the skin from 7-dehydrocholesterol, undergoes two important modifications. The first takes place in the liver through 25-hydroxylation (25(OH)D_3_, calcifediol, or calcidiol). Thereafter, the final conversion to active autocrine hormone 1,25(OH)_2_D_3_ (calcitriol), for the most part, takes place in renal proximal tubular cells by CYP27B1 1α –hydroxylase, a multicomponent/multifunction enzyme in the mitochondria [85,119]. Several vitamin D metabolites are generated in the liver, the kidney, and in other tissue types, which are then excreted in the urine [120,121,122,123]. The most investigated and important forms are 24-hydroxylated (i.e., 24,25-dihydroxyvitamin D3, 1,24,25-trihydroxyvitamin D_3_); these forms are converted from 25-hydroxyvitamin D_3_ and 1,25-dihydroxyvitamin D_3_, respectively. CYP24A1 24-hydroxylase enzyme, which is responsible for these conversions [124,125,126,127]—similarly to CYP27B1, a multicomponent enzyme in the mitochondria—is regulated by calcium, phosphorus, and 1,25-dihydroxyvitamin D3 through the VDR [125,126]. Long-term imbalance in this system or an inappropriate amount of cytochrome P450 enzymes—which control the production, regulation, and degradation of vitamin D—can cause vitamin D insufficiency-related diseases [128]. Thus, abnormally elevated levels of CYP24A1 can create a deficit in vitamin D levels, since this enzyme is uniquely responsible for the catabolism of vitamin D. Elevated levels of CYP24A1 are observed in breast, prostate, esophageal, colon, and lung cancers [129,130], genetically linked hypophosphatemia [131,132], diabetic nephropathy [133,134], and CKD [135,136]. Deficiency of vitamin D can also develop due to an inadequate amount of sun exposure and insufficient nutritional supplementation, or it can appear as a result of certain diseases, e.g., diabetes, cancer, chronic kidney disease (CKD), or genetically linked hypophosphatemia [137].

### 3.1. Effect of Vitamin D and SIRT1 on Metabolic Syndrome

Metabolic syndrome (MetS) is defined with well-determined metabolic alterations [34,35], which serve as a favorable breeding ground and characteristic indicators for many diseases, e.g., obesity, type 2 diabetes (T2DM), CVD, hypertension, cancers, and mental disorders [138,139,140,141,142,143]. It has been shown that vitamin D deficiency is associated with T2DM development [144,145], and its level is negatively correlated with the disease state and positively with insulin sensitivity [146]. We detailed above that vitamin D and SIRT1 have a significant role in a healthy lipid metabolism, as well as in the healthy relationship between metabolic/energy homeostasis and the immune system, where both are linked to regulatory molecules. Indeed, decreased vitamin D is associated with MetS as a whole, and with some components, such as obesity, increased BMI, dyslipidemia, increased blood pressure, and altered insulin and glucose metabolism [25,147]. Thus, vitamin D and SIRT1 may play a beneficial role in the prevention and complementary treatment of metabolic syndrome, since both are key molecules in metabolic/energy sensing and in immune regulation [34,35].

### 3.2. Effect of Vitamin D and SIRT1 on CVD

Vitamin D has important roles in cardiovascular health, as was mentioned above, and, in addition to other risk factors, such as smoking, high cholesterol, hypertension, obesity, and diabetes, vitamin D deficiency is connected to the occurrence of cardiovascular diseases (CVD) [148,149,150]. Although observational studies, preclinical studies, and randomized control trials all showed the beneficial effect of vitamin D on vascular and cardiac functions, linear Mendelian randomization and large clinical trials failed to demonstrate significant benefits on CKD and high-risk CVD populations [151]. However, the vast majority of these trials of vitamin D supplementation did not restrict the study populations to individuals with vitamin D deficiency and did not follow-up the changes of vitamin D level during the trial, correlating the end points with these changes [152]. Additionally, nonlinear Mendelian randomization found significant benefits of vitamin D supplementation on CVD risks [153]. However, the significant positive effect of vitamin D is widely accepted only on the reduction of CVD-related mortality [19,20,23] and all-cause mortality [18,21,22]. The micro-RNA mir34a promotes calcification in vascular smooth muscle cells by downregulation of SIRT1 and Axl [64]. An appropriate level of SIRT1 decelerates vascular calcification by affecting several osteogenic processes, including the reversal of osteogenic phenotypic transdifferentiation in vascular smooth muscle cells, upregulation of eNOS and FoxOs, activation of antioxidant properties, and enhancement of adiponectin release from perivascular adipose tissue [16]. Resveratrol and SIRT1 activators improve cardiovascular health by improving systolic blood pressure and mean arterial pressure, decreasing triglyceride, leptin, inflammation level (Il-6, IL-8, TNFα and CRP) [154], and by maintaining endothelial function and by ameliorating events related to endothelial dysfunction, e.g., impaired vasorelaxation, eNOS uncoupling, leukocyte adhesion, endothelial senescence, and endothelial mesenchymal transition [155].

### 3.3. Effect of Vitamin D and SIRT1 on CKD

Kidney function is indispensable for the synthesis of calcitriol, which allows intestinal calcium absorption in order to maintain extracellular calcium as well as phosphate levels. Therefore, the loss of kidney function leads to elevation of parathyroid hormone (PTH) and, eventually, induces hyperplastic parathyroid growth to re-establish calcium and phosphate balance [156]. The expression, activity, and regulation of 1α-hydroxylase, which is induced by PTH, hypocalcemia, and hypophosphatemia and repressed by FGF23, hyperphosphatemia, hypercalcemia, and calcitriol, is impaired in CKD [157]. The complex of calcidiol and vitamin D-binding protein (25(OH)D_3_/DBP) is filtered through the glomerulus from the circulation and is actively endocytosed into the proximal tubular cells by megalin, a member of the LDL receptor superfamily in the apical membrane [158]. Although PTH is upregulated in CKD, it is insufficient to restore calcitriol synthesis, as the elevated calcium and phosphate levels induce FGF23, metabolic acidosis, and PTH fragments directly inhibiting even non-renal 1α-hydroxylase expression and activity [159]. Additionally, normal kidney function is essential in the maintenance of the serum level of 25(OH)D_3_ for local activation by non-renal 1α-hydroxylase and the autocrine/paracrine actions of the VDR [156]. Since, in CKD, both the glomerular filtration rate (GFR) and megalin content are lower, the amount of ultrafiltrated 25(OH)D_3_/DBP is also lower. However, a sufficient amount of ultrafiltrated 25(OH)D_3_/DBP is required both for its recycling into the circulation to act as a substrate of non-renal conversion and for conversion to calcitriol in the kidney. Therefore, a vicious cycle develops [160,161,162]. Consequently, the correction of the 25(OH)D_3_ deficiency is necessary and sufficient in hemodialysed, and more so in anephric patients, to prevent calcitriol deficiency at an early stage of kidney disease. Moreover, epidemiological studies suggest that vitamin D deficiency is associated more strongly than calcitriol deficiency with a higher risk of disease progression and death in CKD [156]. Of note, the age-dependent decline of renal function observed over the age of 60 resulted in impaired postprandial calcium excretion. Therefore, the daily intake of vitamin D should be considered in light of this physiological decline, in order to prevent vitamin D toxicity in relation to abnormal serum and vascular calcium and phosphate homeostasis [63].

As the liver and the kidney play crucial roles in the maintenance of the active vitamin D level, liver and kidney transplant recipients more frequently develop vitamin D deficiency [163,164]. Additionally, these patients have a higher ratio of vitamin D deficiency-associated comorbidities, such as fractures, diabetes, and infections [165,166]. Since vitamin D regulates the immune system, alteration in its level affects the outcome of allografts, e.g., acute cellular rejection (ACR) and infection [39]. Similarly, several other studies reported that transplanted patients have a higher risk of infections [36,37,167,168,169] and rejections [38,170]. Thus, vitamin D supplementation and tight follow-up of its level is important in transplant recipient patients [39].

### 3.4. Effect of Vitamin D and SIRT1 on Immune System-Related Diseases

1,25(OH)_2_D_3_/VDR signaling is implicated in human innate and adaptive immune response [82,171], but it appears to be species-specific and cell-specific [42]. Based on mouse models, the beneficial effects of vitamin D are suggested in allergies, autoimmune and inflammatory diseases such as inflammatory bowel disease (IBD) [172,173,174], multiple sclerosis (MS) [175,176], autoimmune encephalomyelitis [177], diabetes [178,179,180], systemic lupus erythematosus (SLE) [181], rheumatoid arthritis (RA) [82], and asthma [182]. Prospective observational studies have found that higher vitamin D level is associated with lower rates of T2DM [183], and animal studies suggest that vitamin D promotes β-cell biosynthetic capacity and conversion of proinsulin to insulin [184]. Additionally, vitamin D increases insulin sensitivity, possibly through increased Ca^2+^ influx, stimulated insulin receptor expression, activation of GLUT-4 glucose transporter, and activation of peroxisome proliferator-activated receptor delta (PPAR-δ) [183,185]. However, RCTs failed to prove the positive effect of vitamin D supplementation in risk reduction of T2DM, although patients were not stratified by vitamin D deficiency and the levels of vitamin D were not continuously monitored during the follow-up, similarly to other RCTs in relation to other diseases [186,187]. A meta-analysis of observational studies found significantly decreased vitamin D levels in SLE patients compared to healthy subjects; therefore, vitamin D supplementation with regular monitoring is suggested as part of the health management of SLE patients [181]. Vitamin D induces BCAA catabolism in macrophages and leads to mTOR inhibition; however, parallel to this, it induces the expression of amino acid transporter SLC7A5 in macrophages but not in epithelial cells [42]. SLC7A5 expression level in macrophages positively correlated with clinical parameters and inflammatory conditions in RA patients, and its pharmacological blockade significantly reduced IL-1β level, a downstream target of leucin-mediated mTORC1, the signaling contributor of proinflammatory cytokine production [88]. These findings seem contradictory; however, it is possible that appropriate vitamin D levels simultaneously increase the uptake of amino acids and inhibit mTOR while SLC7A5 then fuels normal anabolic pathways. In contrast, vitamin D in a low level is likely not able to properly coordinate this metabolic network. Additionally, as synovial T cells are relatively insensitive to 1,25(OH_2_)D_3_ compared to circulating blood immune cells, treatment with vitamin D is more potent for enhancing a localized reaction in combination with other RA therapies [188,189]. Moreover, these seemingly contradictory results may have originated in the heterogeneity of this disease; of note, few studies have addressed the analysis of RA subgroups and stages [82]. Although the specific benefits of vitamin D supplementation for treatment and prevention of RA are less accepted because of inconsistent results in randomized clinical trials, conceivable benefits for the improvement of disease of RA, SLE, and osteoarthritis have been reported in meta-analyses [190]. Thus, it is advisable for patients with RA to maintain a serum 25(OH)D_3_ level of at least 30 ng/mL (75 nmol/mL) to prevent osteomalacia, secondary osteoporosis, and fractures [190]. In asthmatic patients, a significant reduction of exacerbation was associated with vitamin D supplementation; this was more pronounced in a subgroup of patients, where vitamin D insufficiency (<30 ng/mL) was diagnosed [182].

### 3.5. Effect of Vitamin D and SIRT1 on Cancer Cells

In relation to cancer cells, vitamin D promoted epithelial differentiation [191] and, consequently, decreased cell proliferation and differentiation in many cancer types, both by direct and indirect pathways [32,192,193]. Vitamin D induced apoptosis [194] and markedly modulated methylation, which leaded to gene repression by histone modification of the DNA [195]. Vitamin D upregulated *p21^WAF/CIP^* and *p27^KIP^* inhibitors of cell cycle arrest in colorectal cancer cells, and since the promoter region of *p21* contains vitamin D response elements, calcitriol can directly regulate *p21* transcription [196]. The promotion of VDR/β-catenin binding reduced the amount of β-catenin binding to T cell factor (TCF), which induces expression of E-cadherin as well as extracellular Wnt inhibitor *DKK-1* [197,198,199]. CYP24A1 mitochondrial protein expression, which determines the half-life of 1,25(OH_2_)D_3_, is regulated both by its methylation status and by vitamin D. Its inhibition facilitates the antiproliferative effect of vitamin D on the downregulation of the WNT/β-catenin pathway and on the inhibition of targeted genes, e.g., *c-Myc*, *TCF1*, and *LEF1* [200]. Vitamin D suppressed antiapoptotic protein expression while inducing proapoptotic protein expression on colorectal cancer cells, but not in normal colon epithelia, where it inhibited proapoptotic signaling by the downregulation of PUMA [199]. Vitamin D diminished the expression of HIF-1, VEGF, as well as IL-8, which are all important angiogenic factors [201]. Vitamin D-induced Nrf2, at a low level, can protect cells from carcinogenic ROS and inflammation, but its constitutive activation caused by mutations or pro-oncogenic signaling can protect cancer cells from cytotoxic effects of chemotherapy, creating chemoresistance [87,199]. Since there are several vitamin D target genes regulating proliferation, migration, invasiveness, differentiation, angiogenesis, extracellular matrix, or immunomodulation of cancer cell, this highlights the possible impact of an appropriate level of vitamin D on its anticancer role [32]. In a meta-analysis of cohort studies, a highly significant linear dose–response relationship was found between the overall survival of breast cancer patient and circulating 25(OH)D_3_ level [21]. Similarly, other prospective and retrospective epidemiological studies reported an association between a 25(OH)D_3_ level below 20 ng/mL and a 30–50% increased risk of colon, prostate, and breast cancer and higher mortality [85]. A systematic review found that vitamin D supplementation induced a shift in colon microbiome composition and increased its diversity [191]. In renal cancer, VDR expression is a prognostic marker and its higher expression predicted a better survival rate [202]. Resveratrol, a natural antioxidant polyphenol and dietary component which upregulates SIRT1, induced apoptosis through activation of p53 by the PI3K pathway, parallel with the inhibition of S6 ribosomal protein in breast cancer cells [203]. This dietary component inhibits estrogen-induced breast carcinogenesis as well through induction of the Nrf2-mediated pathway [204,205]. Moreover, it reverses doxorubicin-chemoresistance by the upregulation of the SIRT1/β-catenin signaling pathway [206]. Studies investigating SIRT1/vitamin D/FOXO interaction suggest a link between VDR, SIRT1, and FOXO3 function, and provide a molecular basis for the cancer chemoprevention actions of 1,25(OH_2_)D_3_ [207,208,209]. Resveratrol suppressed ovarian cancer growth and liver metastasis by inhibiting glycolysis and targeting the AMPK/mTOR pathway [210]. In addition to the beneficial effects of resveratrol on cancer cells, by inducing apoptosis under hypoxia while not affecting normal cells, it also attenuated their migratory properties through downregulation of hypoxia-induced LPA and subsequent activation of HIF-1α and VEGF signaling [92]. Resveratrol, in a cell and organ specific manner, can induce or inhibit hypoxia and ROS production, but both ultimately resulted in cell death in the cancer cells [92]. Additionally, resveratrol suppressed the production of extracellular matrix degrading and remodelling of MMP-2 and MMP-9 [211]. Resveratrol reduces the level of TNF-α, IL-1β, and IL-6 proinflammatory cytokines and suppresses STAT3 and NFκB signaling [212,213]. Moreover, it modulates non-cancer cells in the tumor microenvironment (e.g., CAFs, macrophages, T cells, and endothelial cells), facilitating their tumor-suppressive effects [92]. Resveratrol was suggested as potential alternative to NSAID and selective COX inhibitor in CRC chemoprevention, demonstrating no obvious side effects even after daily oral administration of 5 g/day for 14 days, as reported in a clinical trial [214]. Finally, a dosage ranging from 0–200 µM in combination with FOLFOX (10 µM) was sufficient to enhance antitelomeric and apoptotic potential through resensitization to chemotherapy [87,215].

## 4. Cooperation of Vitamin D and SIRT1 Pathways

Evidence shows a direct and indirect connection of vitamin D and SIRT1, where direct influence exists, on the one hand, through binding of VDR to SIRT1 promoter or to SIRT1 and other proteins in a transcription complex [3,7,13,207,209,216,217,218] and, on the other hand, through epigenetic modifications, by regulating each other [13,32,219]. The indirect routes involve activation of signaling pathways by molecules such as adiponectin, AMPK, resveratrol, methyl-donors, etc., which upregulate/activate either SIRT1 or vitamin D pathways [8,49,96,220,221,222,223,224]. We briefly summarize the possible interaction of vitamin D and SIRT1 in relation to the regulation of signaling pathways and targeted molecules in Figure 2.

### 4.1. Direct Interaction of Vitamin D and SIRT1

#### 4.1.1. Interaction through Binding of VDR to SIRT1 Promoter Region

Ligand-bound VDR can directly bind to *SIRT1* promoter and induce SIRT1 transcription [13]; then, the translated enzyme can activate AMPK [3,7]. This results in reduced fat deposition in skeletal muscle and increased mitochondrial biogenesis and oxidative capacity of muscle fibers [3]. Vitamin D supplementation through SIRT1 upregulation decreased NOX4 expression and ROS production and increased Nrf2 and GLUT4 expression and glucose uptake in high-glucose-treated adipocytes and diabetic mice on a high-fat diet [7]. Additionally, vitamin D supplementation in diabetic mice reduced oxidative stress-induced renal damage through upregulation of SIRT1 and subsequent reduction of NOX4, a NAPH oxidase, which is a characteristic isoform found in the kidney; a high level of NOX4 promotes diabetic nephropathy [95]. Vitamin D deficiency, through decreased SIRT1 level, led to increased fat deposition in adipocytes and macrophage infiltration parallel with increased proinflammatory cytokine IL-6 and TNFα as well as decreased PGC1α, an important molecular partner of the SIRT1/PGC1α antioxidant pathway [12,95].

#### 4.1.2. Interaction through Binding of VDR to SIRT1 Protein or via Epigenetic Modifications of Each Other

VDR can induce expression of both FOXO proteins and SIRT1 and, in a ligand-independent manner, can interact with FOXOs. Additionally, in a ligand-dependent manner, VDR recruits SIRT1 to the VDR/FOXO complex and increase FOXO protein activity and binding to target genes—possibly through deacetylation, similarly to vitamin D regulated SIRT1-induced β-catenin and NFκB deacetylation [207,208].

Vitamin D-sensitive CTCF sites, in the presence of vitamin D, open TADs, which contain SIRT1 promoters and, thus, can activate SIRT1 transcription through epigenetic changes of chromatin and direct binding to the SIRT1 promoter [32,221]. SIRT1 can also modify the activity and DNA-binding capacity of VDR through deacetylation, similarly to epigenetic modifications of other transcription factors [13,219,221].

### 4.2. Indirect Interaction of Vitamin D and SIRT1

Vitamin D can also upregulate SIRT1 protein expression by regulating signaling pathways. Through the upregulation of adiponectin and consequent activation of AMPK, Vitamin D can induce SIRT1 expression [34,35,49]. Cytoplasmic membrane-bound VDR, through activation of SRC, PLA2, PKC, and, consequently, MAPKs, can induce both VDRE and non-VDR target gene transcription, both of which can induce SIRT1 upregulation [43,44,219,221,223]. The antioxidant and antiapoptotic effect of vitamin D is regulated through the activation of the MEK/Erk pathway, which upregulates SIRT1 expression, similar to the process reported in relation to neuronal apoptosis in brain injury, where Erk/SIRT interaction has a neuroprotective role [8]. Vitamin D supplementation inhibited PARP1, a molecule which is important, not only in DNA damage repair, but also in the regulation of cellular stress response, where it blocks SIRT1 function by the depletion of cellular NAD+ level [95]. PARP1 inhibition by vitamin D supplementation had protective effect against diabetic cardiomyopathy, partly through the PARP1/SIRT1/mTOR pathway [217].

## 5. Potential Applications of Vitamin D and Induction of SIRT1 in the Prevention, Treatment, and Reduction of Mortality Risk of Non-Communicable Diseases

### 5.1. Guidelines for Vitamin D Status and Its Daily Intake

As detailed in Table 2, the Institute of Medicine (IOM) and the Endocrine Society provide slightly different guidelines for the classification of vitamin D status based on the blood’s 25(OH)D_3_ level [183].

IOM recommendation for daily intake (RDA) is 600 international units (IUs)/day for individuals 9–70 years of age and 800 IUs/day for those over 70 years of age, with a note that the upper tolerability limit of intake is 4000 IU/day and above that, toxicity may increase [225,226,227]. Although observational studies and meta-analysis reported significant association between vitamin D deficiency and/or supplementation and health status, the interpretation of the results of clinical studies and the ability to draw a well-stratified conclusion is not achievable recently. This is because of the diversity of applied dosage, dosage form, study duration, missing follow-up of vitamin D levels, and polymorphism in vitamin D synthesis, catabolism, DBP, and the VDR, and due to the host potential of epigenetic confounding factors [187]. Additionally, there could be several non-registered influening factors which could affect the sufficiency of vitamin D. For example, methyl donor deficiency can inactivate vitamin D signaling via both disruption of VDR-PGC1α interaction and sequestration of nuclear VDR attributable to HSP90 overexpression [222]. As the determination of the desired serum 25(OH)D_3_ level for a healthy metabolism is still a matter of debate, based on related data, it is advisable to preferably maintain a serum 25(OH)D_3_ level of 30–50 ng/mL (75–125 nmol/L) to achieve the maximum benefits of vitamin D for immune, metabolic, and overall health [190,228,229].

### 5.2. Recommendation for Healthy Individuals to Achieve the 30–50 ng/mL of Serum 25(OH)D_3_ Level

In the temperate zone, a 15 min sunbath, with an uncovered face and four limbs, without any UVB decreasing cream, is accepted as optimal source of vitamin D from April to October. The advantage of this source is that there is no risk of vitamin D toxicity, as the skin stops synthesizing it when the body reaches the desirable level of this hormone. From October to March, vitamin D supplementation is recommended—of note, the skin also produces it, but in a smaller amount than in summertime.

For healthy people without any symptoms of vitamin D deficiency, supplementation is only needed if the serum 25(OH)D_3_ level is lower than 30–50 ng/mL [228,229]. The amount and type of vitamin D as well as the length of supplementation should be discussed and under the control of a physician. However, the daily intake should not exceed 3000–4000 IUs and should allow for complementation by other sources. Of note, the ability of the human body to alternate between different vitamin D sources (i.e., diet, skin, and tissue origin) is lost in a case where the entire required quantity of vitamin D is provided exclusively for long term by dietary supplementation.

### 5.3. Recommendation for Patients with Diseases to Achieve the 30–50 ng/mL of Serum 25(OH)D_3_ Level

Patients with vitamin D deficiency-related diseases should follow the protocol provided by their physicians. There are several guidelines and clinical trials suggesting how to reach and maintain the serum level of 25(OH)D_3_, but it is administered by physicians [228,229,230,231]. In all cases, one needs to consider all conditions which can affect vitamin D production, absorption, and metabolism. Treatments of patient with diseases not related to vitamin D deficiency should consider if the disease affects any areas of vitamin D availability. In all cases, follow-up of the 25(OH)D_3_ level is recommended.

### 5.4. Recommendation to Maintain an Optimal SIRT1 Level in the Body

In relation to SIRT1, both observational studies and clinical trials agree that an appropriate lifestyle and, if needed, SIRT1 inducers, are both beneficial to increase or reset a healthy/healthier homeostasis regulated by the SIRT1 level [34,35].

### 5.5. Recommendation to Preventive and Complementary Treatments

We summarize the potential options for preventive and complementary application of vitamin D from dietary source or supplementation and SIRT1 “inducer” therapies in Table 3.

Prevention has a major role in the maintenance of the well-balanced regulatory network built into the human body. Additionally, there is a possibility, even in imbalanced states, to reset the original homeostasis by the application of evolutionarily inbuilt regulatory mechanisms as complementary treatments; this was summarized in Table 3 from our recent and previous works [34,35]. Continuous overload of any vitamins, including vitamin D, is not recommended, either for prevention or for treatment, except for diseases associated with continuous vitamin deficit. Research and clinical data suggest that vitamin levels that are too high or too low make the body vulnerable to infections and disease development [232,233]. Each vitamin has an optimal level of interval in which it can effectively fulfil its role, without extra energy, regulatory and metabolic processes, for maintaining a healthy homeostasis.

## 6. Conclusions

In our review, we summarized the non-osteometabolic effects of vitamin D and SIRT1 in tissue-specific metabolism and highlighted their common regulatory roles in non-communicable diseases with possible related pathways. We summarized the results of observational studies and clinical trials in order to investigate their potential application in prevention and as a complementary therapy of these types of diseases. In the case of vitamin D, the results of observational studies and clinical trials are not consistent, except in relation to CVD-related mortality. This highlights the importance of well-designed studies in the future. However, our review points to the importance of a complex approach in the prevention and treatment of metabolic alteration- and inflammation-related diseases, as several studies incompletely documented or followed lifestyle habits, such as diet, physical activity, and smoking, or investigated circadian rhythm, which all can influence disease development, progression, and treatment efficacy.

## Figures and Tables

**Figure 1 ijms-24-06154-f001:**
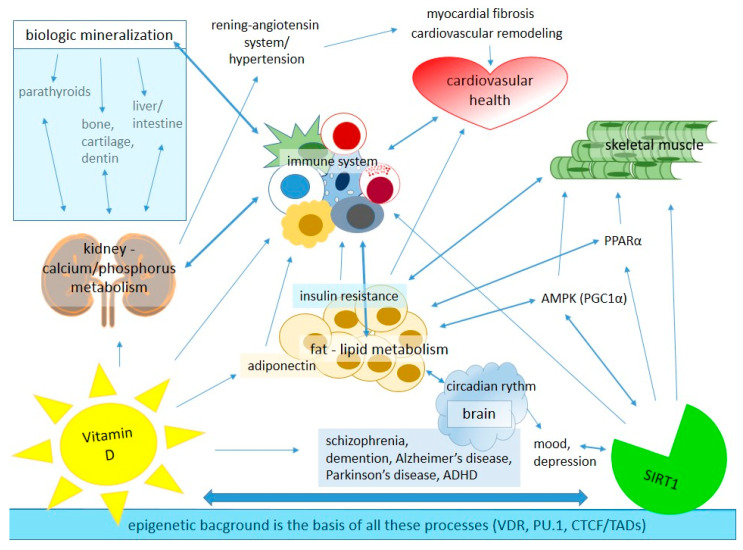
The main roles of vitamin D and SIRT1 in tissue-specific homeostasis in the human body. AMPK: adenosine monophosphate-activated protein kinase, CTCF: chromatin-organizing protein—CCCTC-binding factor, PPAR-α: adenosine monophosphate-activated protein kinase, PU.1: partner pioneer factor, TADs: topologically associated domains, VDR: vitamin D receptor.

**Figure 2 ijms-24-06154-f002:**
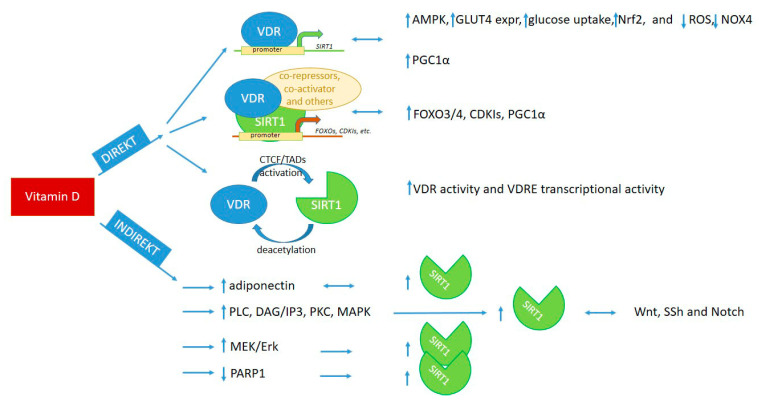
Cooperation of vitamin D and SIRT1 in the regulation of signaling pathways, transcription, and cellular metabolism [3,7,8,12,13,32,43,44,49,87,95,96,97,199,204,205,207,208,209,216,217,218,219,221,222,223]. VDR: vitamin D receptor, VDRE: vitamin D-responsive elements, CTCF: chromatin organizing protein—CCCTC-binding factor, TADs: topologically associated domains.

**Table 1 ijms-24-06154-t001:** Key vitamin D target genes, locations, and function of transcribed proteins in immune cells [77,83,84].

Number	Key Vitamin D Target Genes	Location	Function	Processes	Ref
1	*ACVRL1*	plasma membrane	TGF-β receptor//serine/threonine kinases	signaling	[77]
2	*CD14*	plasma membrane	TLR coreceptor	innate immunity	
3	*CD93*	plasma membrane	intercellular adhesion, clearance of apoptotic cells	innate immunity	
4	*LILRB4*	plasma membrane	leukocyte immunglobulin-like receptor (LIR)	inhibiting immune response	
5	*LRRC25*	membrane bound in the cytoplasm	inhibition of IFN and NFκB signaling pathways	inhibiting immune response	
6	*NINJ1*	plasma membrane	adhesion molecule	inflammation, cell death, axonal growth, cell chemotaxis, and angiogenesis	
7	*SEMA6B*	plasma membrane	axon guidance	NS development	
8	*THBD*	plasma membrane	thrombin receptor	coagulation	
9	*TREM1*	plasma membrane	receptor (Ig superfamily member)	amplifies neutrophil and monocyte-mediated inflammatory responses	
10	*CAMP*	secreted	antimicrobial peptid	innate immunity	
11	*FN1*	plasma membrane, extracellular matrix, secreted into the plasma (blood)	glycoprotein	cell adhesion and migration processes (embryogenesis, wound healing, blood coagulation, host defense, and metastasis)	
12	*SRGN*	secretory granules in hemapoetic cells	secretory granule formation, mediator of cytotoxic cell granule-mediated apoptosis, regulates MMP2 and TNF-α secretion, inhibits bone mineralization	innate and adaptive immunity, Ca^2+^ homesotasis	
13	*CEBPB*	nucleus	transcription factor, regulation of immune and inflammatory response, adipogenesis, gluconeogenesis, liver regeneration, hematopoesis, osteoblast differentiation, osteoclastogenesis	innate and adaptive immunity, metabolism, bone hoeostasis	
14	*THEMIS2*	cytoplasm and nucleus	T cell receptor signaling, regulation of B cell activation, macrophage inflammatory response, promotes LPS-induced TLR4-mediated TNF production	innate and adaptive immunity	
15	*MAPK13*	cytoplasm (member of MAP kinase family, a p38 MAP kinase)	mediates extracellular stimuli; activating ELK1, ATF2 trascription factors, prolactin signaling,	innate and adaptive immunity, proliferation, differentiation, transcription regulation, development	
1	*CYP24A1*	mitochondria	degradation of 1,25-dihydroxyvitamin D3, calcium homeostasis,	vitamin D endocrine system	[83]
2	*G0S2*	mitochondria	positive regulation of extrinsic apoptotic pathway, binding to BCL2	regulation of apoptosis	
3	*HBEGF*	secreted, extracellular space	growth factor	SMC proliferation, cardiac valve formation, normal heart function	
4	*SEMA6B*	type I membrane protein	cell surface repellent	axon guidance	
5	*THBD*	endothelial specific type I membrane receptor	receptor of thrombin	bind thrombin, conversion of protein C to activated protein C	
6	*AQP9*	plasma membrane, multi pass membrane protein	water channel	permeable for glyerol and urea, mediate passage of small, non-charged solutes (carbamides, polyols, purines, pyrimidines)	
7	*CCL7*	secreted	chemotactic factor, attracts moncytes and eosinophils but not neutrophils, binds heparin, CCR1/2/3	attracts macrophages during inflammation and metastasis, augments monocytes antitumor activity, induces release of gelatinase B	
8	*PVALB*	cytoplasm, nucleoplasm	high-affinity calcium binding	muscle relaxation in muscle cells, calcium binding in GABAergic cells	
9	*CD1E*	plasma membrane, MHC-type protein	form heterodimers with beta-2-microglobulin	presentation of lipid and glycolipid antigens to T cells	
10	*NRG1*	plasma membrane, single-pass type I membrane protein	ligand of ERBB3/4 tyrosine kinase receptors	growth, differentiation of epithelial, glial, neuronal and skeletal muscle cells, stimulating milk production, mammary tumor cell differentiation	
1	*CD14*	plasma membrane	TLR co-receptor	innate immunity (independent of the size of the tested target gene set, gene ontology analysis indicates that the modulation of innate immunity is the main physiological outcome of a vitamin D stimulation of human monocytes)	[84]
2	*ORM1*	vesicles in the cytoplasm	acute phase plasma protein		
3	*CAMP*	secreted	antimicrobial peptid		
4	*FBP1*	mitochondria	glucose metabolizing enzyme		
5	*CYP26B1*	endoplasmatic reticulum	Drug- and retinoid metabolism, and synthesis of cholesterol, steroids and other lipids		
6	*TSPAN18*	plasma membrane	unknown		
7–19	+13 other genes	top five biological pathways: neutrophil activation, positive regulation of TNF production, inflammatory response, neutrophil degranulation, negative regulation of T cell proliferation, positive regulation of cytokine secretion			

**Table 2 ijms-24-06154-t002:** Recommendations for the classification of vitamin D status based on blood’s 25(OH)D_3_ level [183]. IOM: Institute of Medicine, ES: Endocrine Society.

25(OH)D_3_ Level (ng/mL)	Recommendation of IOM	Recommendation of ES
lower than 12	deficiency	deficiency
12–19	inadequacy	deficiency
20–29	sufficiency	insufficiency
30–49	sufficiency	sufficiency
higher than 50	reason for concern	sufficiency

**Table 3 ijms-24-06154-t003:** Possible preventive and complementary treatments of non-communicable diseases by targeting vitamin D and SIRT1 level.

Vitamin D	SIRT1	Tissue/Organ Type	Diseases	Signal Pathway/Targeted Molecules	Prevention/Complementary Treatment Options
decrease: BMI, body fat, TGs, CRP, TNFα. increase: SIRT1/6, HDL-C, apo-AI, ABCA1, FFA oxidation	decrease: adipocyte differentiation-PPARγ, inflammation, leptin increase: activates PPAR-α, lipolysis, FFA oxidation, PGC-1α, insulin/mTOR/cell growth, leptin/PI3K-Akt-mTORC1	adipose (white/brown)	T2DM, obesity, insulin resistance	adiponectin/NFĸB, AMPK, PPAR-α	• weight loss • exercise/physical activity • emotional/mental support • nutrition factors (vitamin D, polyphenols, carotinoids, omega-3 FA)
increase: glucose uptake and utilization	increase: FFA oxidation, PGC-1α	skeltal muscle	insulin resistance	adiponectin/NFĸB, AMPK, PPAR-α	
increase: NO production, fat oxidation decrease: blood pressure, cardiac hypertophy, foam cell formation, macrophage activation, vascular inflammation, oxidative stress	decrease: cardiac hypertophy, proinflammatory macrophage activity increase: PPAR-α, eNOS/NO, FOXOs	endothel	cardiovascular abnormalities	adiponectin/NFĸB, AMPK, PPAR-α	
decrease: renin-angiotensin-aldosterone axis		kidney	CKD, hypertension		
decrease: inflammation, proinflammatory cytokines, TNFα and CRP, mTOR, ROS/RNS increase: IL-10, IL-1RA, BCAA catabolism, antioxidant enzymes	decrease: TLR4, IFNγ, IL-1β, IL-6, IL-8 and TNFα levels, oxidative stress, ROS, iNOS, COX-2	immune system		TLR4 Akt/NFĸB/COX-2, Nrf2	
decrease: β-catenin, WNT/β-catenin pathway, c-Myc, TCF1, LEF1, antiapoptotic proteins, HIF-1, VEGF, IL-8 increase: p21WAF/CIP and p27KIP, E-cadherin, DKK-1, proapoptotic proteins, Nrf2	decrease: NFκB, MMP-9, insulin/mTOR/cell growth, leptin/PI3K-Akt-mTORC1, TNF-α, IL-1β, iNOS, COX-2; initiation, promotion, and progression of cancer development increase: ROS, apoptosis, cell cycle arrest	PC, BC, CRC, SkinC	cancer	insulin/mTOR/cell growth, proliferation of leptin/PI3K-Akt-mTORC1	

## Data Availability

Not applicable.

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
