# Peer review of "Interplay of Vitamin D and SIRT1 in Tissue-Specific Metabolism—Potential Roles in Prevention and Treatment of Non-Communicable Diseases Including Cancer"

_ijms, 2023, doi:10.3390/ijms24076154_

Round 1

Reviewer 1 Report

The authors described the interplay between Vitamin D and Sirt1 molecules directly or indirectly. I felt too many volume of Vitamin D explanation, but along with goverment guideline,

the authors delineated the contribution of both Vitamin D and Sirt 1 molecules. I think this manuscript may be suitable for the publication after the minor correction below,

The section of "Cooperation of vitamin D and SIRT1 pathways" should be main topic on this manusctipt, because of the interpretation of both molecule behavior. Indeed, I strongly recommend that the authors should not just draw the hipothetical figure, but also add the detail explantion and discussion in the main part.

Author Response

Authors’ reply to Reviewer1

Thank you for kind you suggestions how to improve the manuscript. We revised the manuscript listed below:

  1. Vitamin D and Sirt1 molecules directly or indirectly. I felt too many volume of Vitamin D explanation, but along with government guideline,

Answer: At the beginning of section 2. we added some sentences (see below) to explain why vitamin D is more discussed than SIRT1 in this section and this review.

„In this section we will detail the role of vitamin D and briefly SIRT1 in healthy metabolic regulation. Vitamin D is discussed in more details and length here since the role of SIRT1 had been already detailed in our previous work, where it easily accessible for those who interested in (34, 35).”

  1. …the authors delineated the contribution of both Vitamin D and Sirt 1 molecules. I think this manuscript may be suitable for the publication after the minor correction below,

The section of "Cooperation of vitamin D and SIRT1 pathways" should be main topic on this manuscript, because of the interpretation of both molecule behaviour. Indeed, I strongly recommend that the authors should not just draw the hypothetical figure, but also add the detail explanation and discussion in the main part.

Answer: We extended with description how vitamin D and SIRT1 cooperate in the metabolism. We added 4.1., 4.1.1., 4.1.2 and 4.2. subsections to section 4.

Additionally, we checked again the text and revised the English if was needed.

All corrections and inserted new parts are labelled as green text in the revised manuscript.

Reviewer 2 Report

1. line 41-42 “included obesity, metabolic syndrome, type 2 diabetes, cardiovascular diseases” : The way of expression is inappropriate. There is a certain mutual inclusion relationship between metabolic syndrome and obesity, type 2 diabetes, cardiovascular diseases.

2. Figure 1 needs to be further beautified to minimize line crossing and text blocking.

3. Figure 1 does not reflect the relationship between epigenetics and others.

4. line 114 SIRT6 was more pronounced: Why is SIRT6 not reflected in Figure 1?

5. line 119-130: There are too many “it”, which lead to unclear meaning. And check in full paper.

6. To further clarify the core idea and make the content more understandable and logical, subtitles should be added.

7. line 307-321: What is the relationship between this passage and resveratrol? Why put “resveratrol” in the title of 3.1?

8. There is no need to put “resveratrol” in the title of 3.1-3.5.

9. The relationship between vitamin D and SIRT1 should be the focus of this article, so the third part should be focused on, but unfortunately this part is quite brief.

10. This review lacks its own views and prospects.

Author Response

Authors’ reply to Reviewer2

Thank you for your suggestions and detailed recommendations to improve the quality of the manuscript. We revised the manuscript listed below:

  1. line 41-42 “included obesity, metabolic syndrome, type 2 diabetes, cardiovascular diseases” : The way of expression is inappropriate. There is a certain mutual inclusion relationship between metabolic syndrome and obesity, type 2 diabetes, cardiovascular diseases.

Answer: We inserted a side sentence „, which otherwise have a certain mutual inclusion relationship as well,” and hope, this could eliminate the probability of inappropriate expression.

  1. Figure 1 needs to be further beautified to minimize line crossing and text blocking.

Answer: We further beautified the Figure 1. We think now it helps more the better understanding and readability of the text.

  1. Figure 1 does not reflect the relationship between epigenetics and others.

Answer: We extend the text related to epigenetics for better understanding the relationship to the processes listed in Figure 1.

  1. line 114 SIRT6 was more pronounced: Why is SIRT6 not reflected in Figure 1?

Answer: The main topic is the SIRT1 and all those information which labelled in Figure 1 is only described and found in relation to SIRT1 and vitamin D. SIRT6 has much less background in the scientific literature in relation to vitamin D and SIRT6 was mentioned from the cited reference only for giving an overall point of view in the text. That is the reason we added to the sentence but would not like to focus on and highlight SIRT6 thus did not added to Fig 1 either.

  1. line 119-130: There are too many “it”, which lead to unclear meaning. And check in full paper.

Answer:

119                  it changed to adiponectin

128-130          “Adiponectin also modulates…”.changed to Functions and phenotypes of macrophages and dendritic cells also modulated by adiponectin via…”

  1. To further clarify the core idea and make the content more understandable and logical, subtitles should be added.

Answer: We added subtitles to all main sections. Newly added subtitles labelled as green text.

  1. line 307-321: What is the relationship between this passage and resveratrol? Why put “resveratrol” in the title of 3.1?

Answer: We deleted resveratrol from the subtitles.

  1. There is no need to put “resveratrol” in the title of 3.1-3.5.

Answer: We deleted resveratrol from 3.1-3.5.

  1. The relationship between vitamin D and SIRT1 should be the focus of this article, so the third part should be focused on, but unfortunately this part is quite brief.

Answer: We added a detailed part of this section based on the references labelled on Figure 2.

  1. This review lacks its own views and prospects.

Answer: We emphasized our own point of view and even added more sentences into the section 4.

Additionally, we checked again the text and revised the English if was needed.

All corrections and inserted new parts are labelled as green text in the revised manuscript.

Reviewer 3 Report

The review has gross errors and is of little relevance to the research.

Author Response

Authors’ reply to Reviewer3

Thank you for your comment.

As there were no detailed explanation which part and approach is thought as error there was no possibility to correct or explain that. The information given this review is based on several other research articles, clinical trials and studies and thus all those research articles’ results and conclusions must be an error. Thank you for your feedback anyway.

Round 2

Reviewer 2 Report

1. line 41-42: What I want to express is “Obesity, type 2 diabetes and cardiovascular disease are one kind of metabolic syndromes”. Therefore, it is inappropriate to juxtapose the concepts of subordination.

2. The conclusion must be simplified. The conclusion only needs to express the core idea of the full text.

Author Response

Authors’ reply to Reviewer2

Thank you for your notes. We modified the manuscript listed below:

Comments and Suggestions for Authors

  1. line 41-42: What I want to express is “Obesity, type 2 diabetes and cardiovascular disease are one kind of metabolic syndromes”. Therefore, it is inappropriate to juxtapose the concepts of subordination.

Answer: In the clinical practice obesity was not a distinct disease in 20 years ago, however, recently diagnosed and treated separately from type 2 diabetes, cardiovascular diseases and cancer. Although metabolic syndrome may involve all the above mention diseases, there are patients with obesity without metabolic alterations as well as type 2 diabetic patients who not even overweight, or there are patients who has metabolic alterations without cancer or any of these other type of diseases. Therefore, it is necessary to clarify these diseases so that what is described in the article is put into a proper context despite of the relationship of these disorders.

Thus, I left all the correction I made in revision one and did not make any new one.

  1. The conclusion must be simplified. The conclusion only needs to express the core idea of the full text.

I deleted some sentences in section 6 and added some other expressions and word, and also changed the section name into “Conclusion”.

Reviewer 3 Report

The article does not meet the requirements of the journal.

Author Response

Authors’ reply to Reviewer3

Thank you for your note. We modified the manuscript listed below:

Comments and Suggestions for Authors

  1. The article does not meet the requirements of the journal.

Answer: We rechecked the requirements of the journal and found that the Tables were not in word format and the vertical lines left in all of them. Therefore, we reformatted Table 1-3, removing vertical lines and replaced the old versions with the new ones in the revised manuscript. Thank you for your notice to improve the manuscript to meet the requirements of the journal.

Round 3

Reviewer 3 Report

The review should be divided into the following points.

1. introduction

2. methods

3. results

4. discussion

5. conclusion

 Please add a methods section for the review. This section should describe the research strategy, including inclusion and exclusion criteria.

Author Response

Authors’ reply to Reviewer3

Thank you for your note. We modified the manuscript listed below:

Comments and Suggestions for Authors

  1. The review should be divided into the following points.
  2. introduction
  3. methods
  4. results
  5. discussion
  6. conclusion

 Please add a methods section for the review. This section should describe the research strategy, including inclusion and exclusion criteria.

Answer: The IJMS Instruction for Authors says: „Review manuscripts should comprise the front matter, literature review sections and the back matter. The template file can also be used to prepare the front and back matter of your review manuscript. It is not necessary to follow the remaining structure.

A review have different structure based on the topic using sections, which allow a the better understanding of the theme. Also a structure of an original article has no meaning in a regular review. Since our review is a regular review, not a systematic review or meta-analysis there are no methods and results sections as usual in this type of review.

Round 4

Reviewer 3 Report

Accepted in the current version.